# Metabotypes of *Pseudomonas aeruginosa* Correlate with Antibiotic Resistance, Virulence and Clinical Outcome in Cystic Fibrosis Chronic Infections

**DOI:** 10.3390/metabo11020063

**Published:** 2021-01-21

**Authors:** Oriane Moyne, Florence Castelli, Dominique J. Bicout, Julien Boccard, Boubou Camara, Benoit Cournoyer, Eric Faudry, Samuel Terrier, Dalil Hannani, Sarah Huot-Marchand, Claire Léger, Max Maurin, Tuan-Dung Ngo, Caroline Plazy, Robert A. Quinn, Ina Attree, François Fenaille, Bertrand Toussaint, Audrey Le Gouëllec

**Affiliations:** 1Département de Biochimie, Faculté de médecine de Grenoble, CNRS, CHU Grenoble Alpes, University Grenoble Alpes, Grenoble INP*, TIMC-IMAG, 38000 Grenoble, France; omoyne@health.ucsd.edu (O.M.); bicout@ill.fr (D.J.B.); dalil.hannani@univ-grenoble-alpes.fr (D.H.); shm@orange.fr (S.H.-M.); leger_claire@orange.fr (C.L.); mmaurin@chu-grenoble.fr (M.M.); cplazy@chu-grenoble.fr (C.P.); btoussaint@chu-grenoble.fr (B.T.); 2Département Médicaments et Technologies pour la Santé (DMTS), University Paris-Saclay, CEA, INRAE, MetaboHUB, 91191 Gif sur Yvette, France; Florence.castelli@cea.fr (F.C.); terrier.samuel@gmail.com (S.T.); francois.fenaille@cea.fr (F.F.); 3Biomathematics and Epidemiology EPSP-TIMC, Veterinary Campus of Lyon, VetAgro Sup, 69280 Marcy l’Etoile, France; 4Laue-Langevin Institute, Theory Group, 71 Avenue des Martyrs, 38042 Grenoble, France; 5Institute of Pharmaceutical Sciences of Western Switzerland, University of Geneva, 1211 Geneva, Switzerland; julien.boccard@unige.ch; 6CHU Grenoble Alpes, Service Hospitalier Universitaire de Pneumologie, Centre de Compétence de la Mucoviscidose, 38000 Grenoble, France; bcamara@who.int; 7Department of Veterinary and biological sciences, Université Claude Bernard Lyon 1, University Lyon 1, VetAgro Sup, UMR Ecologie Microbienne, CNRS 5557, INRA 1418, 69280 Marcy L’Etoile, France; benoit.cournoyer@vetagro-sup.fr; 8CEA, INSERM, CNRS, Bacterial Pathogenesis and Cellular Responses, University Grenoble Alpes, UMR 1036/ERL 5261, 17 avenue des Martyrs, 38054 Grenoble, France; eric.faudry@cea.fr (E.F.); ngotuandung1991@gmail.com (T.-D.N.); ina.attree-delic@cea.fr (I.A.); 9Department of Biochemistry and Molecular Biology, Michigan State University, East Lansing, MI 48824, USA; quinnrob@msu.edu

**Keywords:** cystic fibrosis, metabolomics, multiscale data analysis, LC-HRMS, *P. aeruginosa*, polyamines, Ala-Glu-mesodiaminopimelate

## Abstract

*Pseudomonas aeruginosa* (*P.a*) is one of the most critical antibiotic resistant bacteria in the world and is the most prevalent pathogen in cystic fibrosis (CF), causing chronic lung infections that are considered one of the major causes of mortality in CF patients. Although several studies have contributed to understanding *P.a* within-host adaptive evolution at a genomic level, it is still difficult to establish direct relationships between the observed mutations, expression of clinically relevant phenotypes, and clinical outcomes. Here, we performed a comparative untargeted LC/HRMS-based metabolomics analysis of sequential isolates from chronically infected CF patients to obtain a functional view of *P.a* adaptation. Metabolic profiles were integrated with expression of bacterial phenotypes and clinical measurements following multiscale analysis methods. Our results highlighted significant associations between *P.a* “metabotypes”, expression of antibiotic resistance and virulence phenotypes, and frequency of clinical exacerbations, thus identifying promising biomarkers and therapeutic targets for difficult-to-treat *P.a* infections

## 1. Introduction

Cystic fibrosis (CF) is a severe genetic disorder caused by mutations in the gene encoding the CF transmembrane conductance regulator (CFTR). In the lungs, CFTR ion channel dysfunction triggers impairement of the mucociliary clearance process, which promotes poly-microbial infections [1,2,3]. *Pseudomonas aeruginosa* (*P.a*) is the most frequently isolated pathogen from the sputum of CF adult patients [4,5,6,7]. Highly resistant to antibiotics, *P.a* often causes long-lasting chronic infections responsible for chronic inflammation and subsequent decline of the lung function, as well as episodes of acute exacerbations. *P.a* is thus considered as a leading cause of morbidity and mortality in CF [3,8,9,10,11,12].

Over the years of chronic respiratory infection, *P.a* adapts to this environment and evolves within its host [13,14]. Frequent phenotypic adaptations include acquisition of antibiotic resistances, decreased expression of virulence factors, loss of motility, slower growth, switch to a mucoid phenotype due to overproduction of alginates, and increased formation of biofilm and development of micro-colonies, which all reduce the recognition of the pathogen by the immune system [5,6,15,16,17]. Metabolic changes also occur in response to the nutritional conditions prevailing in CF mucus, such as the emergence of amino acids auxotrophs, likely due to the high cost of metabolic production and ready availability of nutrients in the lung mucus [18,19].

Whole genome sequencing has been useful to identify functional processes triggering *P.a* adaptations to the CF lung [20,21,22]. However, given the complexity of the different levels of regulation of living organisms (post-transcriptional, post-translational, enzymatic kinetics, etc.), the relationships between genome mutations and their effects on relevant phenotypes, such as their resistance to antibiotics or their virulence profile, remain difficult to find [23]. Moreover, recent studies have shown that convergent metabolic adaptations of strains infecting independent patients could be obtained through distinct mutational paths [24] and that isolates with almost identical genome sequences sampled from different patients can express highly divergent transcriptomic, metabolic and phenotypic profiles [25]. Together, these findings highlighted the need to get a more functional view of *P.a*’s within-host evolution, in order to draw links between bacterial adaptations, expression of clinically relevant phenotypes, and ultimate impact of the infection on the patient’s health status. To achieve this goal, metabolomics (that refers to the measure of the small molecules, or metabolites, present in a biological system) constitutes a promising tool [26] as it allows to get a snapshot of bacterial metabolic activities [27]. Comparing internal *P.a* metabolomics profiles obtained during the course of an infection thus provides a functional view of bacterial adaptation to the CF lung environment, at the closest to the phenotype. Precursor studies have shown the potential of metabolomics to study *P.a* metabolic adaptation during chronic infections [18] as well as the link between metabolic profiles and bacterial phenotypes [28,29].

In this paper, we present the first untargeted, non-hypothesis-driven metabolomics study using Liquid Chromatography coupled with High Resolution Mass Spectrometry (LC/HRMS), to access within-host adaptive evolution of *P.a* metabolism within the lungs of chronically infected CF patients. Antibiotic resistance and virulence profiles as well as patients’ clinical health status were also characterized, and the results were integrated with metabolomics footprints in a multiscale statistical approach. This strategy allowed us to define bacterial metabolic profiles that we named “metabotypes” that are significantly associated with clinically relevant bacterial phenotypes and to the patients’ respiratory disease. As such, our results demonstrate the potential of untargeted metabolomics to get insights into bacterial adaptation processes that can be connected to pathogenicity and clinical outcome. This study thus constitutes a first and important step in the identification of future metabolites that could be used as biomarkers and metabolic targets for next-generation therapies to support the clinical care of these difficult-to-treat infections.

## 2. Results

### 2.1. Evolutionary Relationships of P.a Clinical Isolates

In order to study *P.a* metabolism over the course of chronic CF lung infections, a retrospective longitudinal collection of *P.a* clinical isolates sampled from expectorations of chronically infected adult CF patients was built. Thirty-four patients were included into the study (clinical description of the cohort in Appendix A). For each patient, 3 to 5 isolates sampled at different time points of the 2010–2015 follow-up period were arbitrarily selected for Pulsed-Field Gel Electrophoresis (PFGE) of SpeI-restricted total DNA genotyping (Figure 1a). Evolutionary clonal lines were defined as isolates sampled from the same patient at different time points but sharing PFGE profiles that clustered into a same clonal complex (CC) as defined by Römling et al. [30,31] (Figure 1b). This analysis identified clonal evolutionary lines for 32/34 patients, confirming the chronic nature of *P.a* infection (Figure 1c). Only two patients (17 and 86) have been excluded from further analysis due to no detection of an evolutive clonal line. To note, one patient (patient 27) appeared to be co-infected by two distinct clonal lines, and both have been included in the study. We also observed 3 CCs (CC1, 2 and 3) shared between different patients. From this analysis, a Final Cell Bank was built by pairing the earliest and the latest isolates from each of the 33 clonal lines (same CC). They represent different evolutionary stages (hereafter referred to as early and late) of the within-host adaptation of the clone which had initiated the chronic infection in the past before the first sample collection.

### 2.2. Acquisition of P.a Metabolomic Profiles by Untargeted LC-HRMS

The isolates of the Final Cell Bank were analyzed by untargeted LC/HRMS to determine their intracellular metabolic content (Figure 2a). Isolates were first grown in synthetic CF medium 2 (SCFM2), without mucin, which mimics the nutritional conditions in CF pulmonary mucus ([32], see Methods). Intracellular extracts of mid-log cultures were harvested by fast-filtration and mechanical lysis, and the contents were normalized by considering the ratio between colony-forming units (CFU) over optical density at 595nm (OD_595_) (CFU/OD_595_) as indicated in Aros-Calt et al. [33]. Metabolomic analysis was performed using two complementary LC/HRMS methods to maximize chemical coverage, and data were processed and normalized following the most up-to-date methods (see Methods). Comparison of spectral data with public and in-house databases allowed annotation of 271 metabolites (Appendix A). Intensities of these 271 putatively annotated metabolites were analyzed using multivariate statistical methods described in the following sections. Statistically significant metabolites were finally formally identified by matching their tandem MS/MS fragmentation profiles with a standard or by manual interpretation (see Methods, Figure 2a, Appendix A).

### 2.3. Diversity of P.a Metabolic Evolution within CF Patients’ Lungs

To investigate intra-host modifications of *P.a* metabolic profiles during the course of CF chronic lung infections, we computed *metabolic Polarity degrees_i,j_* (*P_i,j_*) as an indicator of the modifications of the metabolomic signatures between early and late isolates (see Methods). *Metabolic P_i,j_* allows to distinguish the “core metabolome” of each clonal line, which remains unchanged over time (*P_i,j_* close to 0), from the “variable metabolome”, which varies during infection (*P_i,j_* ≠ 0) (Figure 2b). Metabolic *P_i,j_* were computed to describe the within-host changes in the production of the 271 annotated metabolites measured by LC-HRMS over our 33 evolutionary lines (Figure 2c). Statistical analysis of the metabolic *P_i,j_* showed the existence of several within-host evolutionary paths of the *P.a* metabolomes, highlighting the diversity of the metabolic adaptations (Appendix A). 

### 2.4. Intra-Host Metabolic Adaptation Is Associated with the Acquisition of Antibiotic Resistance

Given that acquisition of antibiotic resistances is a hallmark of chronic infection and is of major clinical importance in CF lung infections, we next investigated the within-host modifications of *P.a* metabolomic profiles associated with concomitant antibiotic resistance changes. Resistance profiles of each *P.a* isolate against 14 antibiotics of clinical importance were determined in vitro (Appendix A). *P_i,j_* computation was then applied to this phenotypic dataset to capture within-host modifications of the resistance phenotypes *j* for each clonal line *i* and identified either gain, loss or unchanged resistance phenotype between early and late isolates. Hierarchical cluster analysis (HCA) showed a segregation of the clonal lines according to the overall gain, loss or unchanged antibiotics resistance profiles (row clustering, Figure 2d). Interestingly, resistance profiles to the beta-lactams family as well as to the two aminoglycosides gentamicin and amikacin evolved in a similar manner within the patient’s airways (column clustering, Figure 2d). 

This antibiotic resistance *P_i,j_* was then integrated with the previously described metabolic *P_i,j_* following a multiscale statistical workflow (Figure 2e). Briefly, we performed a hierarchical clustering on principal components (HCPC) inferred from a multiple factor analysis (MFA) to jointly extract the information from both metabolomics and antibiotic resistance datasets. This analysis led to the identification of 3 clusters representing *P.a* clonal lines with similar “metabo-resistome” modifications between early and late isolates. Cluster 1 (Figure 2e, in red) was significantly associated with the acquisition of resistance for 11/14 of the tested antibiotics (of which 9/9 beta-lactams), and with intra-host modifications of 5/271 annotated metabolites. These modifications included increased levels of 13-Hydroxyoctadeca-9,11-dienoic acid, methyl-4-hydroxyphenylacetate, Ala-Glu-meso-diaminopimelate and decreased levels of both isobutyric and diaminopimelic acids (Figure 2e). Clusters 2 and 3 were associated with an unchanged or decreased antibiotic resistance, respectively (in white and blue, Figure 2e). 

The 5 metabolites associated with Cluster 1 were selected to build a supervised logistic model, in order to predict acquisition of beta-lactams resistance from a reduced number of metabolite modifications (Figure 2e). Step-by-step forward selection and internal cross-validation of the best model showed that identification of clonal lines that had gained beta-lactam resistance over time could be predicted based on changes in their metabolome. More specifically, the sole increase in Ala-Glu-meso-diaminopimelate production between early and late isolates predicted the acquisition of beta-lactams resistance with a moderate sensitivity (67%) but excellent specificity (92%) (Figure 2f).

### 2.5. P.a Metabotypes Segregated by Differential Levels of Polyamines and Their Metabolites

In order to highlight potential metabolic signatures of clinical relevance, we also analyzed *P.a* metabolic profiles, without integrating the temporality of the samplings (Figure 3a). Sixty-six *P.a* isolates were described by 2 data blocks: LC-HRMS intensities of 151 selected metabolites (Figure 3b,c,f left side), and presence/absence of 6 virulence phenotypes (Figure 3d–f right side)

For the metabolite analysis, metabolites presenting differential expression levels within the bank (variation coefficient >0.5), which are most likely associated with clinically relevant bacterial phenotypes, were selected. HCPC analysis of the 151/271 selected metabolites led to the identification of 3 groups of *P.a* isolates with similar metabolic profiles, which will hereafter be termed metabotypes (Figure 3b). The 10 most significant metabolites associated with these 3 metabotypes are listed in Table 1 (one-way analysis of variance (ANOVA) F-test *p*-value <0.05). In order to put these data into a biological perspective, metabolic pathways associated with these metabolites were inferred using the PAMDB database [34].

Interestingly, metabotype 1 was characterized by significantly lower levels of spermidine and putrescine, nucleotides, nucleosides, hexosamines or precursors of glycosaminoglycan (AMP, ADP, CDP, GDP, cytosine, guanosine, UDP-galactose and UDP-N-acetylgalactosamine). Metabotype 2 was characterized by higher levels of the ornithine catabolite N2-succinylornithine. Finally, metabotype 3 was characterized by higher levels of the ornithine precursor N-acetyl-L-ornithine and lower levels of the ornithine catabolite N2-succinylglutamate-semialdehyde. Metabotype 3 was also characterized by a higher level of 1-hydroxy-2-nonyl-4(1H)-quinolinone (NQNO) (Table 1).

These results highlighted metabolites involved in the polyamines (e.g., putrescine, spermidine) metabolism as strongly discriminant between the various *P.a* metabotypes (Figure 3c,g, pairwise Student *t* test, FDR adjusted <0.05, except for spermidine level between metabotypes 2 and 3, *p*-value = 0.076).

### 2.6. Multivariate-Based Analysis of Bacterial Virulence

To access the relationships between *P.a* metabotypes and virulence properties of each isolate, we investigated six different phenotypes: cytotoxicity on macrophages and epithelial cells, epithelial cells stress response induced by *P.a* infection (see Appendix A), formation of mucoid colonies, pigment production and bacterial growth rate. Experimental results were converted into a binary presence/absence matrix and analyzed in a multivariate fashion by HCPC analysis. HCPC clustering revealed 3 groups of isolates that can be defined according to their relative level of virulence (Figure 3d) as follows: (i) avirulent isolates (*n* = 30/66), which are generally non-cytotoxic on J774 macrophages and A549 epithelial cells, do not cause stress on A549, and have a slow growth rate; (ii) moderately virulent isolates (*n* = 17/66), which cause stress on A549, are cytotoxic on J774 but not on A549, and have a fast growth rate; and (iii) highly virulent isolates (*n* = 19/66), which are cytotoxic on both cell types and cause stress on A549 cells (χ2 *p*-values <0.05, Figure 3e).

### 2.7. Polyamines Production Is Associated with the Level of P.a Virulence

Significant associations between established virulence levels and metabotypes were observed: metabotype 1 matched with avirulent isolates; metabotype 2, with moderately virulent isolates; and metabotype 3, with highly virulent isolates (Fisher exact test *p*-value = 0.01, Figure 3f). Considering that the polyamines’ metabolic pathway was among the most discriminant between the different metabotypes, the relationships between the virulence phenotypes and the intensity of putrescine and spermidine were further investigated. Firstly, the relative changes in the proportion of these two metabolites were found to be highly correlated (Pearson’s R = 0.91, *p*-value <2.10^−16^), confirming the link between the production of spermidine and the level of its precursor putrescine and therefore the modifications of the metabolic flux related to this pathway between metabotypes. Secondly, analysis of the relationships between the level of these 2 metabolites and virulence properties revealed that spermidine levels were significantly higher in the isolates which were cytotoxic on J774 macrophages and significantly lower in the isolates forming mucoid colonies. Moreover, spermidine and putrescine levels were higher in fast-growing isolates (*p*-value of Student *t*-tests <0.05). Although not statistically significant, a trend toward a gradual increase of polyamines’ production was detected in the bacterial isolates with higher virulence (Figure 3g).

We then analyzed the modifications of *P.a* virulence level occurring overtime within the patients’ airway. The overall level of virulence was lowered between early and late isolate for 9/33 clonal lines (27%) (for example, with the early isolate being defined as highly virulent and the late isolate defined as moderately or non-virulent; see Figure 3d–f), increased for 3/33 clonal lines (9%), and no change was observed for the remaining 21/33 clonal lines (64%). Interestingly, Kruskal–Wallis testing showed a significant decrease in the spermidine level between early and late isolates, when the virulence level decreased (*p*-value of Kruskal–Wallis test = 0.03). A similar trend was observed for putrescine, although not significant (*p*-value of Kruskal–Wallis test = 0.11) (Appendix A).

### 2.8. High Polyamines Production By P.a Is Associated with Frequent Clinical Exacerbations

The patients’ clinical records were used to build clinical indicators of respiratory health, in order to explore the relationships between *P.a* metabotypes and clinical outcome. Temporal monitoring of the Forced Expiratory Volume in 1 s (FEV1) and comparisons to a reference population are considered the most robust prognosis predictor in CF patients [8,36,37]. Three indicators describing the average level (high or low), long-term (decline or no decline) and short-term (stable or unstable) dynamics of the patients’ lung functions (see Methods, Figure 4a) were thus built based on all the FEV1 measurements performed during the 2010–2015 period (5–45 measures per patient). No statistical relationship was observed between these 3 indicators, confirming that they were non-redundant describers of the patient’s respiratory health status. Remarkably, a significant association between production of both putrescine and spermidine by the cultivated *P.a* isolates, and the short-term dynamic of the patient’s respiratory function (Wilcoxon test, *p*-values = 0.030 and 0.041 for associations between an unstable FEV1 and the levels of putrescine and spermidine, respectively, Figure 4b) was observed. These significant relationships showed, for the first time, that metabolite biomarkers specifically produced by *P.a* are correlated to the frequency of acute clinical exacerbations of the patient’s respiratory illness. Considering the statistical inferences of the results described in Figure 3, this relationship could be related to the expression of virulence factors in *P.a* isolates. 

## 3. Discussion

Preliminary genome-based studies gave important insights into *P.a* patho-adaptation during CF chronic lung infections, but gene-to-phenotype relationships are, to date, still difficult to draw. The concept of this study was based on the hypothesis that downstream metabolic manifestations induced by genome-based adaptations would be more readily linkable to bacterial phenotypes and their clinical impact. A multi-factorial investigation based on the comparison of *P.a* population genetic structures, intracellular metabolic profiles, and patients’ health records was built. Longitudinal clinical *P.a* isolates, representative of different evolutionary stages of a clonal complex chronically infecting a patient’s airway, were collected and used to build the reference clonal evolutionary lines studied here. Early and late isolates of each clonal evolutionary line were extensively characterized by the analysis of virulence and antibiotic resistance properties and the acquisition of untargeted and high-resolution in vitro metabolomic fingerprints.

Pairing metabolic profiles from sequential *P.a* isolates of a same evolutionary line (through the calculation of *P_i,j_*) allowed the differentiation between the “core” and the “variable” metabolomes of each *P.a* line. The core metabolome made of metabolites whose production remained unchanged over time (*P_i,j_* close to 0), was discarded from the dataset, thus giving a specific emphasis to metabolites showing significant changes over time. This “variable metabolome” was made of a lower number of metabolites whose production was strongly correlated to bacterial phenotypes of clinical importance. These changes were associated with modulations of bacterial metabolic pathways in response to within-host selective pressures.

In particular, the sole increase of Ala-Glu-meso-diaminopimelate production between early and late *P.a* isolates was found to be sufficient to predict within-host acquired beta-lactam resistance. This murein tripeptide is known to be produced during the degradation of the bacterial cell wall peptidoglycan and directly reused in the recycling process [38]. Interestingly, the therapeutic target of beta-lactam antibiotics is primary peptidoglycan synthesis. Our results clearly support the hypothesis that the increase in Ala-Glu-meso-diaminopimelate production reveals an over-activation of the peptidoglycan recycling process, thus allowing the bacteria to escape from the effect of beta-lactam treatment. Recently, inactivation of the peptidoglycan recycling pathway has been shown to be associated with restoration of antibiotic sensitivity, decrease in bacterial virulence, and improvement of the innate immune system response in vitro [39,40]. Our observations support these earlier observations and underline the importance of this pathway in the acquisition of *P.a* beta-lactam resistances in vivo. Peptidoglycan recycling inhibitors thus represent promising targets for future antimicrobials. Our results also suggest that Ala-Glu-meso-diaminopimelate concentration can be used as a biomarker to anticipate the efficacy of beta-lactam antibiotic treatments: An increased level in Ala-Glu-meso-diaminopimelate can indeed predict the acquisition of beta-lactams resistance with moderate sensitivity (67%) but excellent specificity (92%). We could then anticipate that such metabolic readouts can be measured directly from the patients’ expectorations and provide a fast and relevant information, which could advantageously complement routinely used bacterial culture methods (antibiograms), which require 24 to 48 h of growth in conditions far from the patients’ airways. Although Ala-D-Glu-meso-diaminopimelate levels would not detect 100% of beta-lactams resistances, high levels of the metabolite would clearly indicate that a beta-lactam treatment would most likely fail to control the infection. This would thus provide fast and highly valuable information to redirect the clinical team towards the choice of a different antibiotic therapy and avoid ineffective try-and-fail therapeutic cycles. A clinical research protocol should be conducted to validate that measurements of Ala-D-Glu-meso-diaminopimelate in the sputum of CF patients will contribute to the early diagnosis of beta-lactam resistance and to a more rational use of antibiotics.

On the other hand, cross-sectional analysis of *P.a*’s untargeted metabolomics footprints allowed us to classify the isolates according to their metabolic profiles and highlighted the polyamines pathway as a major discriminant between the different metabotypes. These molecules are found in all living organisms and are notably involved in promoting cell growth [41,42]. This could explain the concordance between the levels of metabolites involved in polyamines and nucleic acids synthesis. The main polyamines found in bacteria are putrescine, spermidine and cadaverine [43,44]. As a major result, significant correlations were found between a high production of putrescine and spermidine by *P.a* isolates, the expression of several virulence phenotypes, as well as an unstable pulmonary function. It should be noted that highly unstable pulmonary function mostly refers to particularly severe clinical phenotype of “frequent exacerbations. Interestingly, it has been shown in vitro that *P.a* possesses the operons *spuABCDEFGH-spuI* for polyamines uptake and utilization. It was also shown that the global CbrAB two-component system senses polyamines, regulates the *spu* operons and modulates 236 genes which have effect on metabolism, virulence and antibioresistance in *P.a* [45,46]. In another study, Zhou et al. demonstrated that spermidine has a positive effect on the activation of the Type 3 Secretion System (T3SS) and found a relation with cytotoxicity on epithelial cells in *P.a* laboratory strains [47]. Furthermore, Twomey et al. reported an increased level of putrescine in bronchial secretions of CF patients during pulmonary exacerbations. This increase was correlated with the presence and abundance of *P.a* and *Chrysiogenales*, but the producers of these molecules remained to be defined [48]. As such, our results draw a link between these two studies, demonstrating for the first time the link between high levels of putrescine and spermidine specifically produced by *P.a*, high *P.a* virulence and clinical exacerbations.

Putrescine is also known to be a precursor for the synthesis of succinate [49] and was found to be a biomarker of acute *P.a* lung infection in a pre-clinical model close to CF exacerbations [50]. Moreover, Whiteson and colleagues have shown that putrescine produced by clinical *P.a* isolates induces the activation of human dendritic cells and the production of interleukin (IL)-12, leading to a pro-inflammatory polarization of the immune response (CD4 T cells) [51] that could in turn lead to lung inflammation and subsequent clinical exacerbation episodes and loss of lung function. Future work should address the role of polyamines production by *P.a* in the crosstalk with host cells. Our results suggest that a significant increase of polyamines production by *P.a* could induce a boost of bacterial growth and virulence, inducing an important inflammatory response and subsequent acute exacerbation. Such findings direct future developments for CF treatments towards a better control of polyamines production. Promising results have recently been reported using specific antibodies preventing the integration of extracellular polyamines by *P.a*, thus reducing T3SS expression, in vitro cytotoxicity against A549 cells, as well as in vivo mortality in an animal infection model [52].

Finally, and although it is often dominant, *P.a* is not unique but is one of the bacteria that make up the lung microbiota of CF patients. There are many interactions between the species composing the CF lung microbiota, and they can have a significant impact on the clinical outcome of the infection [53]. Thus, if phenotypic studies on pure isolates such as those presented here provide important elements for understanding the patho-evolutionary mechanisms of chronic *P.a* infection, the hypotheses raised need to be studied in the context of the CF pulmonary ecosystem as a whole. For example, it would be of great interest to study the influence of isolates of *P.a* isolates producing high levels of polyamines on the structure of the microbiota and the impact of the microbial community on host cells. Indeed, several human pathogens possess transport systems allowing the use of extracellular polyamines to support growth [42]. It is thus possible that the overproduction of polyamines by particularly virulent isolates of *P.a* isolates induces the overgrowth of other colonizing species in the patients’ lungs, thus causing the dysbiosis at the origin of the exacerbations. For this, microbiota culture models under conditions reproducing the CF lung environment could be used, in order to assess the impact of *P.a* producing different polyamines levels on the diversity and abundance of different species [54].

In conclusion, the reported datasets demonstrate that non-targeted metabolomics is an efficient strategy to identify bacterial mechanisms of clinical importance, bringing out potential novel therapeutic strategies. Monitoring of metabolites found among the flexible metabolome might be used to predict exacerbations or resistance to certain antibiotic therapies. Such biomarkers could help in rationalizing the use of antibiotics and provide alternatives or supplements to conventional antibiotic therapies, ultimately improving patients’ health care.

## 4. Materials and Methods

### 4.1. Patients

#### 4.1.1. Cohort Selection of Patients

Thirty-four patients from the Cystic Fibrosis Resource and Competence Center (CFRCC) of the Grenoble-Alpes University Hospital (CHUGA) were recruited retrospectively according to the following inclusion criteria: diagnosis of CF according to the national guidelines, 16 years old in 2010, chronically infected by *P.a* according to the EuroCareCF reference criteria [55], 3 *P.a* isolates sampled over a 3 years follow-up period. The data are derived from clinical research on cystic fibrosis patient and bacterial samples named METAPYO: “A metabolomic approach for the study of the adaptive evolution of *Pseudomonas aeruginosa* during chronic pulmonary infections in cystic fibrosis”. This research was approved by the CHUGA institutional review board and authorized after its filing with the CNIL according to the french procedure for a monocentric retrospective study (Reference Methodology MR004-Compliance Commitment No. 2205066 v 0). Duly informed patients did not object to the conduct of the research.

#### 4.1.2. Clinical Data and Respiratory Function Modelling

All the Forced Expiratory Volume in 1 s (FEV1) measurements made between 2010 and 2015 have been extracted from the medical records of the cohort’s patients. FEV1 values (in L) have been converted in Z-score adjusted for age, sex and height by non-linear regression following the Global Lung Initiative recommendations. These international guidelines also advocate to adjust this Z-score according to ethnicity, but the French legislation does not allow to collect this information. Considering the greatest prevalence of CF in Caucasians, we therefore considered all our patients as of Caucasian origin in this calculation [56].

Patients have been stratified according to both the level and the temporal dynamics of all FEV1 records during the 2010–2015 follow-up period. Mean FEV1 was classified as high if above the cohort average, low if not. Long-term dynamic of the FEV1 was defined as in decline if the slope of the linear regression of the FEV1 over 5 years was significantly below 0, not declining if not. Short-term dynamic was defined as unstable if the standard deviation of the residuals around the linear regression of the FEV1 over time was above 40%, stable if not. One indicator of each Mean FEV1, Long-term dynamic and Short-term dynamic have been used to describe each patient’s clinical state over the study period.

### 4.2. P.a Clinical Isolates

#### 4.2.1. *P.a* Isolates Identification

*P.a* isolates from sputum of CF patients followed at Grenoble-Alpes University Hospital were obtained from the Grenoble-Alpes University Hospital Microbiology Laboratory. Strains have been isolated and purified according to the national guidelines [57]. Isolates were stored at −80 °C in cryotubes with beads. *P.a* isolates were identified by standard biochemical testing and proteomic profiling by matrix-assisted laser desorption and ionization time-of-flight mass spectrometry (MALDI-TOF MS) (Bruker Daltonics, Wissenbourg, France). Three to 5 isolates per patient (see cohort selection section) were arbitrarily selected and cultivated in SCFM2 medium for further analysis.

#### 4.2.2. Growth Conditions

If not specified, all pre cultures and cultures were done at 37 °C, 230 rpm, in SCFM2 medium in aerobic conditions. Pre-cultures were done in 2.5mL SCFM2, cultures in 2 ∗ 2.5 mL SCFM2, pooled together prior to experimental procedure. SCFM2 medium was prepared as indicated by Turner et al. [32]. Mucin was discarded from the medium composition in order to allow a precise follow-up of bacterial growth by Optical Density measurements at 595 nm (OD_595_).

#### 4.2.3. Pulsed-Field Gel Electrophoresis Clonal Analysis

PFGE analysis of *SpeI* restricted genomic DNA was performed as described previously [58]. PFGE was performed using a CHEF-DR III apparatus (Bio-Rad), set at 5.0 V/cm, with a linear ramping from 5 to 25 s for 11 h and 5 to 60 s for 13 h. PFGE gels were pictured using ImageLab 5.1 software (Bio-Rad). Images were then aligned and analyzed using BioNumerics software (version 7.1, Applied Maths, Sint-Martens-Latem, Belgium) DNA patterns (or pulsotypes) were converted into a 0/1 discrete matrix of presence/absence of bands at each molecular weight, as described previously by Lavenir et al. [59]. Hamming’s distances have been calculated using R software version 3.3.2 [60] (Vienna, Austria), and the number of different bands between isolates have been interpreted following the criteria defined in Römling et al. (1995) [31]. Two or more pulsotypes sharing less than 7 different bands have been defined as a PFGE Clonal Complex (CC), attesting for a recent common ancestor.

### 4.3. Metabolomics Analysis

#### 4.3.1. Sample Preparation

Sampling and metabolite extraction of *P.a* isolates grown in SCFM2 were performed as indicated by Aros-Calt and colleagues, with slight modifications [33,61] (See Supplementary Methods). Bacterial culture, sample preparation, metabolomics analyses, and data processing were performed in biological triplicates.

#### 4.3.2. Liquid Chromatography Coupled with High Resolution Mass Spectrometry (LC-HRMS) Analysis

Untargeted metabolomic profiling of the bacterial samples was done using ultra high-performance liquid chromatography (Ultimate 3000 UPLC, Thermo Fisher Scientific, Waltham, MA, USA) coupled with an Exactive Orbitrap mass spectrometer (Thermo Fisher Scientific Waltham, MA, USA). In order to enhance the chemical coverage of the analysis, we used two different but complementary chromatographic columns, consisting in reversed phase chromatography (C18 chromatographic column) and Hydrophilic Interaction Liquid Chromatography (HILIC) for the analysis of hydrophobic and polar metabolites, respectively.

The C18 chromatographic separation was carried out on a Hypersil GOLD C18 column (1.9 µm, 150 × 2.1 mm, Thermo Fisher Scientific) at 30 °C, with flow elution rate of 500 μL/min. The mobile phases consisted of A (100% water + 0.1% formic acid) and B (100% acetonitrile (ACN) + 0.1% formic acid). Elution started with an isocratic step of 2 min at 5% mobile phase B, followed by a linear gradient from 5% to 100% mobile phase B for the next 11 min. These proportions were kept constant for the next 12.5 min before returning to 5% B for 4.5 min. The HILIC chromatographic separation was carried out on a Sequant ZIC-pHILIC column (5 µm, 150 × 2.1 mm, Merck, Darmstadt, Germany) maintained at 15 °C under a elution gradient of mobile phases A and B at a flow elution rate of 200 μL/min. Mobile phase A was 10 mM ammonium carbonate pH 10.5 (adjusted with ammonium hydroxide), and mobile phase B was 100% ACN. Elution was initiated with 80% B phase for 2 min, followed by a linear gradient of 80–40% B from 2 to 12 min. The chromatographic system was then rinsed for 5 min at 0% B, before returning at 80% B and the and the run ended with an equilibration step of 25 min at 80% B.

The mass spectrometer was fitted with an electrospray source (ESI) operating in positive and negative ionization modes for C18 and ZIC-pHILIC, respectively. It was operated with capillary voltage at −3kV in the negative ionization mode and 5 kV in the positive ionization and a capillary temperature set at 280 °C. Temperature of the autosampler compartment was set at 4 °C, and the injection volume was 10 μL. Detection was carried out from *m*/*z* 75 to 1000 in both ionization modes at a resolution of 50,000 at *m*/*z* 200 as reported by Aros-Calt et al. [61] (each scan taking 0.5 s).

#### 4.3.3. LC-HRMS Data Processing

Raw LC-HRMS data were converted to m/z extensible markup language (.mzXML) in centroid mode using MSConvert ProteoWizard (release version 3.0.9393). Peak detection and integration were performed using R version 3.3.2 and XCMS package version 3.0.2 [62]. Briefly, features were detected using the centWave algorithm (step = 0.01, m/z deviation tolerance = 10 ppm, peak width = 10–40 s for C18, 20–120 s for HILIC, signal-to-noise ratio = 5). Peaks were grouped by density and retention times were nonlinearly smoothed (loess). Missing values (gap filling) were imputed by the chrom method. Annotation of adducts, fragments, and isotopes was achieved using the CAMERA package [63].

Features detected following XCMS-CAMERA analysis were then filtered and standardized using the Workflow4Metabolomics platform [64,65]. Data filtering was done according to the following criteria: (i) correlation coefficient between dilution factor and peak area in QC samples > 0.7, (ii) ratio of mean peak area in blanks over biological samples < 0.33, and (iii) variation coefficient of peak area in QC samples <30%. Peak intensities were then normalized using the Probabilistic Quotient Normalization (PQN) algorithm described by Dieterle et al. [66].

#### 4.3.4. Metabolite Annotation

Feature annotation was performed by using our spectral database first according to accurate measured masses and chromatographic retention times [61,67,68] and then according to publicly available databases KEGG, PAMDB, HMDB, and METLIN [34,69,70,71] solely using accurate masses. This data-based analysis allowed putative annotation of 271 metabolites (Appendix A). Metabolite identification was further confirmed for discriminant metabolites LC-MS/MS experiments using a Dionex Ultimate chromatographic system combined with a Q-Exactive mass spectrometer (Thermo Fisher Scientific, San Jose, CA, USA) under non-resonant collision-induced dissociation conditions using higher-energy C-trap dissociation (HCD). To be identified, metabolites had to match at least two orthogonal criteria (among accurate measured mass, retention time, and MS/MS spectrum) to those of an authentic chemical standard analyzed under the same analytical conditions, as proposed by the Metabolomics Standards Initiative [72]. In the absence of an available authentic chemical standard, metabolites of interest were only considered as putatively annotated based on accurately measured masses and interpretation of the MS/MS spectra when available as described by Aros-Calt et al. 2015 [61]. Under these conditions, up to 51 discriminant metabolites were characterized: 30 had accurate masses, retention times and MS/MS matching those of an authentic standard, 3 were putatively annotated by matching their MS/MS spectra to those from the METLIN public database or showed MS/MS spectra consistent with both the proposed structures and the spectra of structural homologues, 1 shared accurate mass and retention time with an authentic standard, and 17 compounds were only annotated based on their accurate masses (Appendix A).

### 4.4. Phenotypic Assays

Cytotoxic potential of the *P.a* clinical isolates on eukaryotic cells was tested according to previously described protocols [73,74], with slight modifications. Antibiotic resistance phenotypes were tested following the recommendations of the European Committee for Antimicrobial Susceptibility Testing 2017 criteria [75]. All phenotypic assays are detailed in supplementary Methods.

### 4.5. Polarity Degree_i,j_

To investigate intra-host modifications of *P.a* phenotypic and metabolic profiles during the course of CF chronic lung infections, an indicator, the Polarity degree_i,j_
*(P_i,j_)* of the relative differences in metabolite production between early and late isolate of each evolutionary line infecting a patient’s airway, was computed as follows:(1)$$P_{i,j}=\frac{L_{i,j}-E_{i,j}}{L_{i,j}+E_{i,j}};V/\mathrm{cm}$$
where *L_i,j_* and *E_i,j_* stand for late and early isolates intensities for evolutionary line i, metabolite *j*, respectively. *P_i,j_* thus returns a value in the interval [−1,1], with positive/negative values representing an increase/decrease in intensity of metabolite *j* in line *i* overtime.

### 4.6. Statistical Analyses

All statistical analyses were performed in R software version 3.3.2 (R Core Team, Vienna, Austria). Multivariate statistical analysis, notably Hierarchical Clustering on Principal Components (HCPC, [76]) analyses, were performed using FactoMineR R package [77], and graphical representations have been done using factoextra [78].

Most of the multivariate statistical analyses presented in this work rely on the HCPC method published by Husson et al. [76]. Briefly, the HCPC algorithm is divided into 3 steps. First, the dimensions are reduced by a factorial method, such as a Principal Component Analysis (PCA) for quantitative variables, a Multiple Correspondence Analysis (MCA) for categorical data, or a Multiple Factorial Analysis (MFA) to jointly integrate different data blocks [79]. Second, a Hierarchical Cluster Analysis (HCA, ward method, Euclidean distances) is performed on the components to determine groups of samples or individuals sharing similar profiles. The optimal number of clusters was calculated by analyzing the gain in inertia provided by the addition of a new group (default parameters, as described in [77]). Finally, a k-means partition [80] is applied to stabilize the previous HCA classification.

#### 4.6.1. Multiscale Integration of within-Host Adaptation of Antibiotic Resistance and Metabolomics Profiles

In order to identify metabolites predictive of acquired antibiotic resistance, we designed a multi-scale statistical workflow. First, we calculated the *P_i,j_* representing the within-host modifications of both metabolite intensities and antibiotic resistances between early and late isolates of each evolutionary line. Then, we conducted a multiscale unsupervised HCPC based on MFA to extract the common information from the two blocks of metabolite and antibiotic resistance *P_i,_*_j_. The output of the HCPC analysis was then used to select variables (metabolites and antibiotic resistance phenotypes) found as statistically associated. Finally, we built a supervised logistic model based on the selected variables, in order to predict the acquisition of antibiotic resistance phenotypes from the modifications of a minimum number of metabolites intensities. The best model was selected by step-by-step forward analysis based on the Akaike information criterion and validated by internal cross validation.

#### 4.6.2. Definition of Bacterial Metabotypes

Variable selection of the most differentially expressed metabolites (i.e., most likely to be associated with differential phenotype expression) was performed (151/271 putatively annotated metabolites with a variation coefficient >0.5). Bacterial metabotypes were defined by HCPC analysis based on MFA, with metabolite intensities spread over two blocks, according to the method that allowed the metabolite detection (C18 or HILIC), in order to balance the influence of each block on the final PCs.

#### 4.6.3. Definition of Bacterial Level of Virulence

HCPC analysis was performed on the bacterial phenotypes (cytotoxicity against A549 and J774, stress induced on A549, growth speed, pigment production, and mucoidy) coded into binary classes. Analysis of variable categories associated with each cluster allowed us to define the bacterial level of virulence.

## Figures and Tables

**Figure 1 metabolites-11-00063-f001:**
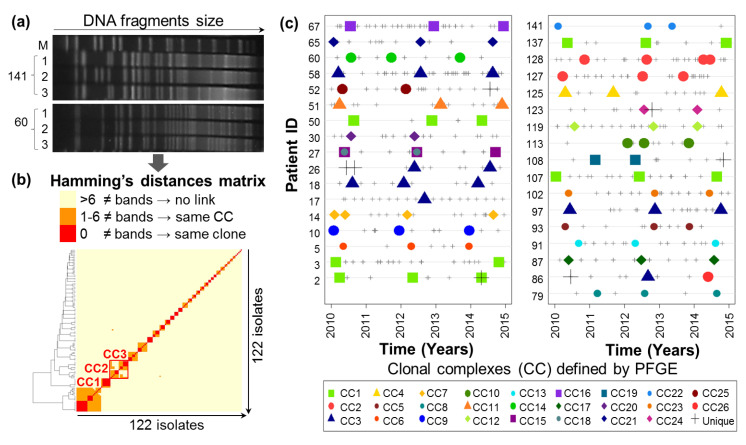
Temporal collection and Pulsed-Field Gel Electrophoresis (PFGE) genotyping of clinical *P.a* isolates responsible for chronic lung infection in CF patients. (**a**) Representative example of PFGE gels (pulsotypes) of *P.a* clinical isolates sampled from patients 60 and 141 (1, 2 and 3 represent the sampling time of each isolate: beginning, middle or end of the 2010–2015 follow-up period, respectively; M = marker fragment size). (**b**) Pairwise comparison of 122 *P.a* PFGE pulsotypes. Clonal complexes (CCs) and clones have been defined according to Römling et al. criteria [31]: no clonal link if the pulsotypes showed more than 6 different bands, Clonal Complex (CC) if the pulsotypes showed less than 6 different bands, and clonal if the pulsotypes were identical. (**c**) Temporal series of PFGE-genotyped isolates sampling from patients’ expectorations between 2010 and 2015. Small grey crosses represent all the *P.a.* isolation time points between 2010 and 2015 for each patient. The symbol shape of the bigger dots represents the CC of the isolates that have been analyzed by PFGE (unique clones are represented by a black cross). For 32/34 patients, the earliest and latest sampled isolates belonging to a same CC have been selected for the Final Cell Bank.

**Figure 2 metabolites-11-00063-f002:**
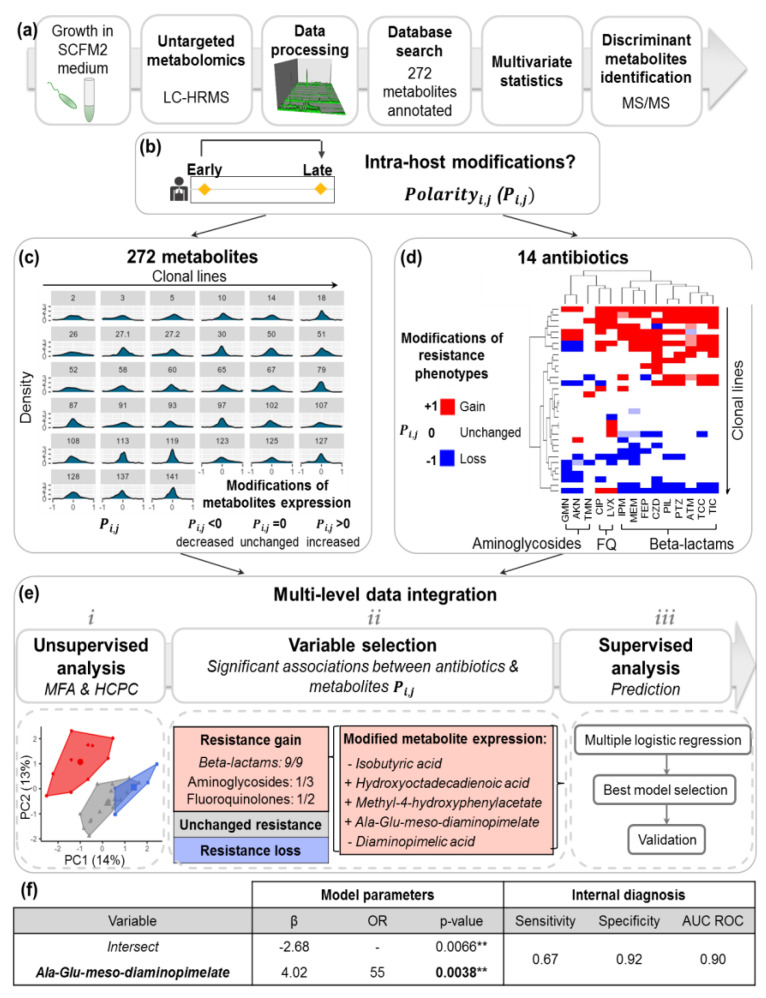
Multiscale analysis identifies within-host metabolic modifications of *P.a* associated with the acquisition of beta-lactam resistances. (**a**) Schematic summary of untargeted LC-HRMS metabolomic analysis workflow for acquisition of *P.a* isolates’ metabolic profiles. (**b**) Intra-host modifications between early and late isolates from 33 evolutive lines have been assessed by calculating the *P_i,j_* value for each line *i*, metabolite or resistance phenotype *j*. (**c**) Distributions of metabolite P_i,j_ representing intra-host modifications of 271 annotated metabolite intensities. (**d**) Hierarchical cluster analysis of antibiotic resistance P_i,j_ representing intra-host modifications of resistance against 14 antibiotics (*). (**e**) Multiscale data integration workflow for the identification of metabolic signatures associated with the acquisition of antibiotic resistance: (*i*) unsupervised HCPC identified 3 clusters of *P.a* lines with similar modifications of both antibiotic resistances and metabolite intensities; (*ii*) cluster 1 (in red) is significantly associated with acquisition of antibiotic resistances, especially against beta-lactams (χ2 *p*-value < 0.05), and with modifications in the abundance of 5 metabolites; (*iii*) selection of significantly associated metabolites and antibiotics to build a supervised logistic model predicting acquisition of antibiotic resistances from metabolic changes; (**f**) cross-validation of the best logistic regression model, which predicts the acquisition of beta-lactam resistance from an increased production of Ala-Glu-meso-diaminopimelate. ** *p*-values < 0.01. (*) abbreviations: FQ: fluoroquinolones; AKN: amikacin, ATM: aztreonam, CIP: ciprofloxacin, CZD: ceftazidime, FEP: cefepime, GMN: gentamicin, IPM: imipenem, LVX: levofloxacin, MEM: meropenem, PIL: piperacillin, PTZ: piperacillin-tazobactam, TCC: ticarcillin-clavulanic acid, TIC: ticarcillin, TMN: tobramycin.

**Figure 3 metabolites-11-00063-f003:**
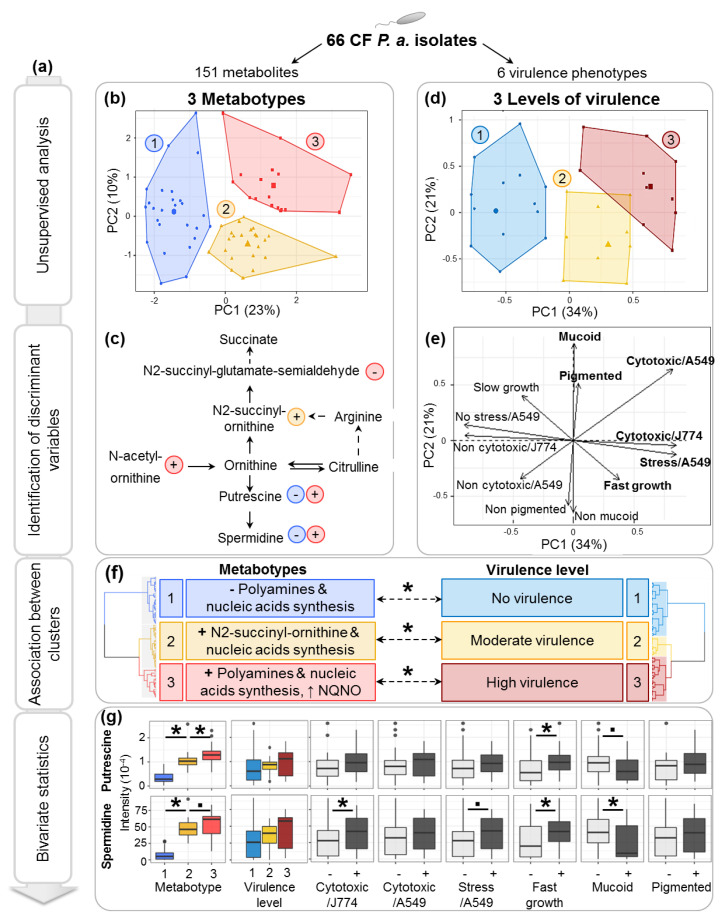
Multiscale analysis identified associations between *P.a* metabotypes and expression of virulence phenotypes. (**a**) Summary of multiscale statistical workflow. (**b**) Representation of 3 HCPC clusters of isolates expressing similar metabolic profiles (metabotypes) in the first 2 PCs of the MFA. Cluster centroids are shown as larger dots. (**c**) Simplified view of the polyamines (putrescine, spermidine) synthesis pathway (adapted from [35]). Significant metabolites are highlighted by arrows, indicating whether the metabolite is found in higher or lower abundance in the color-matching metabotype. (**d**) Representation of 3 HCPC clusters of isolates expressing similar virulence phenotypes in the first 2 PCs of Multiple Correspondence Analysis (MCA). Clusters centroids are shown as larger dots. (**e**) Representation of the virulence categories in the first 2 PCs of the MCA. (**f**) Dendrograms and summary of the main characteristics of HCPC clusters identified on each data block and associations between *P.a* virulence level and metabotype (Fisher test *p*-values <0.05 are indicated by (*)). (**g**) Boxplots showing bivariate relationships between putrescine and spermidine levels with the expression of individual virulence factors (Student *t* test *p*-values <0.1 and <0.05 are indicated by (.) and (*), respectively).

**Figure 4 metabolites-11-00063-f004:**
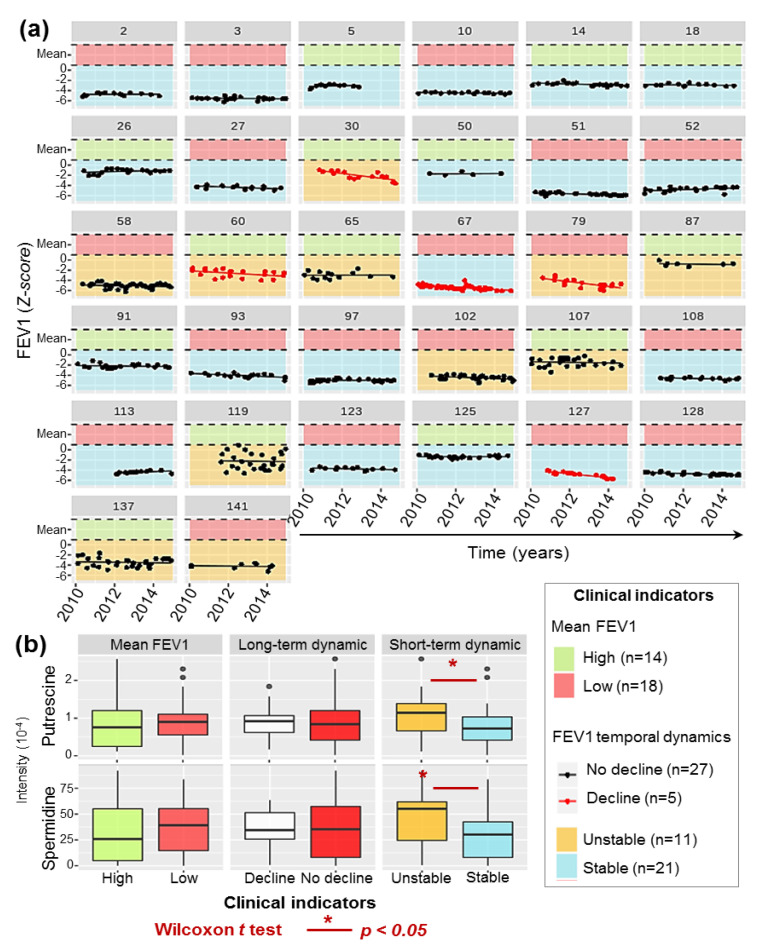
Relations between patients’ respiratory functions and the production of polyamines by *P.a* isolates. (**a**) Temporal series of all FEV1 measurements realized on the 32 patients of the cohort during the follow-up period and construction of clinical indicators. Average FEV1 for each patient was defined as high (green background) or low (red background), by comparison to the cohort average. FEV1 temporal dynamics was modeled using linear regression of the FEV1 measurements overtime as follows: (i) long-term dynamic was classified as in decline (red lines) if the slope was significantly under 0, not declining (black lines) if not; (ii) short-term dynamics were defined as unstable (yellow background) if the standard deviation of the residuals was above 40% or stable (blue background) if not. (**b**) Boxplots showing the putrescine and spermidine production (LC-HRMS peak intensity) by *P.a* per FEV1 based respiratory categories. The data presented in this figure have been translated into a table to make it accessible to colorblind readers (Appendix A).

**Table 1 metabolites-11-00063-t001:** List of the 10 Metabolites Most Strongly Associated with the Three Metabotypes Observed Among the *P.a* Clinical Isolates.

Metabotype	Metabolite	Relative Abundance	*p*-Value (One-Way ANOVA)	Identification Status (***)
**1**	Spermidine	−	2.8 × 10^−11^	a, c, d
	Cytosine	−	5.2 × 10^−10^	a, b, d
	Putrescine	−	1.3 × 10^−0.9^	a, c, d
	Adenosine monophosphate (AMP)	−	3.6 × 10^−0.9^	a, b, d
	Uridine diphosphate (UDP)-Galactose (UDP-Glucose)	−	4.6 × 10^−0.9^	a, b, d
	Cytidine diphosphate (CDP)	−	5.9 × 10^−0.9^	a
	Adenosine diphosphate (ADP)	−	8.1 × 10^−0.9^	a, b, d
	Guanosine	−	1.9 × 10^−0.8^	a, c, d
	N2-Succinyl-L-ornithine	−	2.6 × 10^−0.8^	a
	UDP-N-acetylgalactosamine (UDP-N-acetylglucosamine)	−	4.1 × 10^−0.8^	a, b, d
**2**	Guanine	+	1.4 × 10^−0.6^	a, b, d
	UDP-N-acetylgalactosamine (UDP-N-acetylglucosamine)	+	1.8 × 10^−0.6^	a, b, d
	12-Hydroxydodecanoic acid	+	2.6 × 10^−0.6^	a, b, d
	Guanosine monophosphate	+	4.5 × 10^−0.6^	a, c, d
	Pentoses phosphate	+	1.7 × 10^−0.5^	a, b, d
	N2-Succinyl-L-ornithine	+	2.3 × 10^−0.5^	a
	Glucosamine 6-phosphate (Galactosamine 6-phosphate)	+	2.4 × 10^−0.5^	a, b, d
	Cytosine	+	3.7 × 10^−0.5^	a, b, d
	Guanosine	+	5.2 × 10^−0.5^	a, c, d
	UDP-Galactose (UDP-Glucose)	+	1.5 × 10^−0.4^	a, b, d
**3**	1-Hydroxy-2-nonyl-4(1H)-quinolinone	+	7.5 × 10^−0.8^	a
	Palmitoleic acid	+	2.5 × 10^−0.7^	a
	Glycerylphosphorylethanolamine/ sn-glycero-3- phosphoethanolamine	+	3.0 × 10^−0.7^	a, f
	AMP	+	2.2 × 10^−0.6^	a, b, d
	N2-Succinyl-L-glutamic acid 5-semialdehyde	−	3.3 × 10^−0.6^	a
	Heptadecenoic acid	+	3.6 × 10^−0.6^	a
	N-Acetylornithine	+	5.4 × 10^−0.6^	a, c, d
	Tetradecanoyl-phosphate (n-C14:0)	+	1.0 × 10^−0.5^	a
	Indoleglycerol phosphate	+	1.2 × 10^−0.5^	a
	Glycerol	+	1.2 × 10^−0.5^	a

(*) Identification status: (a) based on accurate mass, (b) based on ZIC-pHILIC column retention time similarity with a standard, (c) based on C18 column retention time similarity with a standard, (d) based on MS/MS spectrum similarity with a standard, (e) based on the MS/MS spectra similarity with those from the METLIN public database, (f) based on the MS² spectra. 1, 2 and 3 in the table mean respectively metabotype 1, 2 and 3.

## Data Availability

Mass Spectrometry data. LC-HRMS data collected from the 198 bacterial cultures (66 *P.a* isolates x 3 biological replicates) can be found on MASSIVE repository under the following identifiers: MSV000084790 (C18 pos) and MSV000084791 (HILIC neg).

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
