# Peer review of "Metabotypes of Pseudomonas aeruginosa Correlate with Antibiotic Resistance, Virulence and Clinical Outcome in Cystic Fibrosis Chronic Infections"

_metabolites, 2021, doi:10.3390/metabo11020063_

Round 1

Reviewer 1 Report

The manuscript addresses a very interesting study regarding a comparative metabolomics analysis of isolates from chronically infected cystic fibrosis patients in order to provide a functional view of P. aeruginosa adaptation. Authors integrated metabolic profiles with expression of bacterial phenotypes and clinical measurement and performed multiscale analysis.

Authors presented a nice and well-presented work; I have no specific comments and accept the manuscript in present form. 

Author Response

We thank you for your time and appreciate that you have no specific comments on the manuscript in its current form.

Reviewer 2 Report

The paper is well written and structured. The Introduction provide a full overview of the topic, and points out the most remarkable issues concerning the research area. The approach is innovative, and the auhtors were very able to provide a high lelev of scientific soundness to the manuscript.

Methods are adequately presented, and results properly described. Perhaps, the only suggestion I would like to give is to address some considerations included in the results, which could be moved to the discussions, as the results section is exclusively aimed at reporting the data obtained.

The Figures are clear and exaustive.

Author Response

We agreed that some sentences did not belong in the results section and we deleted them or moved them to the discussion section.

We have made the following changes:

-We have deleted "suggesting acquisition or loss of cross resistance mechanisms" lines 166-167 from the results section.

- We have deleted the sentence “Interestingly, this metabolite is known to be involved in the recycling process of the peptidoglycan cell wall, which is the primary target of beta-lactam antibiotics [34]” lines 190-191. This information was already in the discussion part lines 397-402.

- We have deleted “Polyamines are known to be involved in DNA replication, gene expression and protein synthesis. They can also act as growth factors [36–38], which could explain the concordance between the levels of metabolites involved in polyamines and nucleic acids synthesis (Table 1).” lines 239-242 and we have added the sentence “This could explain the concordance between the levels of metabolites involved in polyamines and nucleic acids synthesis.” in the discussion part, lines 429-430.

- We have deleted “This is consistent with previous studies that showed a tendency of P.a to lower the expression of virulence factors during its adaptation to CF airways [5,16,17].” Lines 302-304.

- We have deleted “Overall, these data showed significant and consistent relationships between putrescine and spermidine production, and the expression of P.a virulence-associated phenotypes.” lines 308-310 ever discussed in the discussion part.

-We have deleted “It should be noted that highly unstable pulmonary function mostly refers to particularly severe clinical phenotype of "frequent exacerbations" lines 321-322 and moved it to the discussion part lines 433-435.

Reviewer 3 Report

The manuscript ID: metabolites-1070031 by Moyne and colleagues describes the use of an untargeted, non-hypothesis-driven metabolomic approach to identify significant metabolites correlating with Pseudomonas aeruginosa adaptation processes during Cystic Fibrosis (CF) chronic lung infection. The authors managed to describe a direct correlation between Ala-Glu-meso-diaminopimelate production and the development of resistance to β-lactams during the pathogen adaptation to the CF lung environment, and between the production of polyamines and the exhibition of a more virulent behaviour, even corresponded by exacerbation episodes in the patients. Thus, the authors propose the metabolomic analysis as a novel and valuable tool to predict P. aeruginosa population dynamics and adaptation within the hosts in CF lung infection.

The manuscript is well structured and clear. The methods and results are rigorously detailed, well supported by statistical analysis and well commented in the discussion section. The described data are promising and interesting, especially in the context of a culture-independent analysis of CF sputum samples aimed to better characterize P. aeruginosa infection dynamics.  However, although supported by a robust statistical analysis, such results should be confirmed in further hypothesis-driven studies, characterized by a specific strain selection, before being considered as predictive markers of pulmonary exacerbations or antibiotic resistance development. This should be better specified in the conclusions.

Moreover, was there a specific reason for excluding the P. aeruginosa strains from patients 17 and 86 rather than comparing their features to those of isolates belonging to the same clonal complexes (CC2 and CC3) from other patients?

Overall, after minor revisions, the paper can be considered suitable for publication in “Metabolites”.

MINOR COMMENTS

-Please avoid any comment in the Results section and move them to the Discussion section; specifically, move or integrate lines 166-167, 190-191, 239-242, 302-304, 307-309 and 319-322 into discussion;

-Please define the symbol ** in Figure 2F caption.

Author Response

Response to Reviewer 3 Comments

The manuscript ID: metabolites-1070031 by Moyne and colleagues describes the use of an untargeted, non-hypothesis-driven metabolomic approach to identify significant metabolites correlating with Pseudomonas aeruginosa adaptation processes during Cystic Fibrosis (CF) chronic lung infection. The authors managed to describe a direct correlation between Ala-Glu-meso-diaminopimelate production and the development of resistance to β-lactams during the pathogen adaptation to the CF lung environment, and between the production of polyamines and the exhibition of a more virulent behaviour, even corresponded by exacerbation episodes in the patients. Thus, the authors propose the metabolomic analysis as a novel and valuable tool to predict P. aeruginosa population dynamics and adaptation within the hosts in CF lung infection.

The manuscript is well structured and clear. The methods and results are rigorously detailed, well supported by statistical analysis and well commented in the discussion section. The described data are promising and interesting, especially in the context of a culture-independent analysis of CF sputum samples aimed to better characterize P. aeruginosa infection dynamics. However, although supported by a robust statistical analysis, such results should be confirmed in further hypothesis-driven studies, characterized by a specific strain selection, before being considered as predictive markers of pulmonary exacerbations or antibiotic resistance development. This should be better specified in the conclusions.

We have modified our conclusions in accordance with this remark. We have reworded it as follow lines 421-424 "A clinical research protocol should be conducted to validate that measurements of Ala-D-Glu-meso-diaminopimelate in the sputum of CF patients will contribute to the early diagnosis of beta-lactam resistance and to a more rational use of antibiotics" and at the same place we have deleted the sentence “Routine Ala-D-Glu-meso-diaminopimelate measurements will be implemented in our clinical unit to prospectively validate this observation in our CF cohort. If validated, this monitoring will efficiently contribute to a more rational use of antibiotics, and reduce the spread of multidrug resistant bacteria”.

Moreover, was there a specific reason for excluding the P. aeruginosa strains from patients 17 and 86 rather than comparing their features to those of isolates belonging to the same clonal complexes (CC2 and CC3) from other patients?

As explained in lines 104 to 116 of the revised manuscript, our goal was to develop clonal evolutionary lines. We therefore decided to eliminate patients 17 and 86 because we wanted to study the evolution of metabolism in relation to the evolution of virulence, resistance and clinical data. Therefore, in order to be able to calculate the degrees of polarity, we selected early and late isolates for each patient, which was not the case for patients 17 and 86.

Overall, after minor revisions, the paper can be considered suitable for publication in “Metabolites”.

MINOR COMMENTS

-Please avoid any comment in the Results section and move them to the Discussion section; specifically, move or integrate lines 166-167, 190-191, 239-242, 302-304, 307-309 and 319-322 into discussion;

Responses :

We have taken the remarks into account and made the following changes:

-We have deleted "suggesting acquisition or loss of cross resistance mechanisms" lines 166-167 from the results section.

- We have deleted the sentence “Interestingly, this metabolite is known to be involved in the recycling process of the peptidoglycan cell wall, which is the primary target of beta-lactam antibiotics [34]” lines 190-191. This information was already in the discussion part lines 397-402.

- We have deleted “Polyamines are known to be involved in DNA replication, gene expression and protein synthesis. They can also act as growth factors [36–38], which could explain the concordance between the levels of metabolites involved in polyamines and nucleic acids synthesis (Table 1).” lines 239-242 and we have added the sentence “This could explain the concordance between the levels of metabolites involved in polyamines and nucleic acids synthesis.” in the discussion lines 429-430.

- We have deleted “This is consistent with previous studies that showed a tendency of P.a to lower the expression of virulence factors during its adaptation to CF airways [5,16,17].” Lines 302-304.

- We have deleted “Overall, these data showed significant and consistent relationships between putrescine and spermidine production, and the expression of P.a virulence-associated phenotypes.” lines 308-310 ever discussed in the discussion part.

-We have deleted “It should be noted that highly unstable pulmonary function mostly refers to particularly severe clinical phenotype of "frequent exacerbations" lines 321-322 and moved it to the discussion part lines 433-435.

As the deletion of all these sentence have an impact on the numeration of bibliography we updated the bibliography.

-Please define the symbol ** in Figure 2F caption.

In figure 2F, we added the signification of the symbol ** as follow line 211 “(f) cross-validation of the best logistic regression model, which predicts the acquisition of beta-lactam resistance from an increased production of Ala-Glu-meso-diaminopimelate. ** p-values <0.01”